# D-Lactic Acid Production from Sugarcane Bagasse by Genetically Engineered *Saccharomyces cerevisiae*

**DOI:** 10.3390/jof8080816

**Published:** 2022-08-03

**Authors:** Warasirin Sornlek, Kittapong Sae-Tang, Akaraphol Watcharawipas, Sriwan Wongwisansri, Sutipa Tanapongpipat, Lily Eurwilaichtr, Verawat Champreda, Weerawat Runguphan, Peter J. Schaap, Vitor A. P. Martins dos Santos

**Affiliations:** 1National Center for Genetic Engineering and Biotechnology (BIOTEC), 113 Thailand Science Park, Phahonyothin Road, Pathum Thani 12120, Thailand; warasirin@biotec.or.th (W.S.); kittapong@biotec.or.th (K.S.-T.); sriwan@biotec.or.th (S.W.); sutipa@biotec.or.th (S.T.); weerawat.run@biotec.or.th (W.R.); 2The Laboratory of Systems and Synthetic Biology, Wageningen University and Research, Stippeneng 4, 6708 WE Wageningen, The Netherlands; peter.schaap@wur.nl; 3Department of Microbiology, Faculty of Science, Mahidol University, 272 Rama VI Road, Ratchathewi, Bangkok 10400, Thailand; akaraphol.wat@mahidol.ac.th; 4National Science and Technology Development Agency, 111 Thailand Science Park, Phahonyothin Road, Pathum Thani 12120, Thailand; lily@nstda.or.th; 5Laboratory of Bioprocess Engineering, Droevendaalsesteeg 1, 6708 PB Wageningen, The Netherlands; 6LifeGlimmer GmbH, Markelstrasse, 38, 12163 Berlin, Germany

**Keywords:** D-lactic acid, *Saccharomyces cerevisiae*, hybrid yeast, glycerol-3-phosphate dehydrogenase genes, *gpd* deletion, *adh* deletion

## Abstract

Lactic acid (LA) is a promising bio-based chemical that has broad applications in food, nutraceutical, and bioplastic industries. However, production of the D-form of LA (D-LA) from fermentative organisms is lacking. In this study, *Saccharomyces cerevisiae* harboring the D-lactate dehydrogenase (DLDH) gene from *Leuconostoc mesenteroides* was constructed (CEN.PK2_DLDH). To increase D-LA production, the CRISPR/Cas12a system was used for the deletion of *gpd1*, *gpd2*, and *adh1* to minimize glycerol and ethanol production. Although an improved D-LA titer was observed for both CEN.PK2_DLDHΔ*gpd* and CEN.PK2_DLDHΔ*gpd*Δ*adh1*, growth impairment was observed. To enhance the D-LA productivity, CEN.PK2_DLDHΔ*gpd* was crossed with the weak acid-tolerant *S. cerevisiae* BCC39850. The isolated hybrid2 showed a maximum D-LA concentration of 23.41 ± 1.65 g/L, equivalent to the improvement in productivity and yield by 2.2 and 1.5 folds, respectively. The simultaneous saccharification and fermentation using alkaline pretreated sugarcane bagasse by the hybrid2 led to an improved D-LA conversion yield on both the washed solid and whole slurry (0.33 and 0.24 g/g glucan). Our findings show the exploitation of natural yeast diversity and the potential strategy of gene editing combined with conventional breeding on improving the performance of *S. cerevisiae* for the production of industrially potent products.

## 1. Introduction

Environmentally sustainable platforms for the production of commodity and specialty chemicals are needed to replace current fossil-based platforms. Carbon-neutral renewable resources, including lignocellulosic waste streams, are seen as particularly promising for the environmentally sustainable production of bio-based chemicals. Lactic acid is a bio-based chemical used in the food, chemical, and healthcare industries. Lactic acid is used to make polylactic acid (PLA), a biodegradable polymer that is widely used in packaging material, mulch film, garbage bags, and medical materials [1,2]. PLA is typically produced from racemic mixtures, although it can also be produced from pure enantiomers to make homopolymers of poly-L-lactic acid (PLLA) or poly-D-lactic acid (PDLA). However, the physical properties of racemic PLA, i.e., brittleness and low thermostability, still limit its use as a replacement for conventional plastics. One way to improve the physical properties of PLA is to create a stereocomplex structure comprising PLLA and PDLA in different ratios. The stereocopolymer has varying physical properties, including a higher melting point, depending on the ratio of the stereoisomers, which broadens the applications [3].

PLA stereocopolymers are synthesized by the polymerization of high-purity L-lactic acid (L-LA) and D-lactic acid (D-LA). The most commercially available LA is the L-form. A variety of microorganisms, including bacteria, fungi, yeast, cyanobacteria, and algae, have been reported for their ability to produce LA; however, most produce LA as racemates. The development of microbial strains for the robust production of stereoisometrically pure LA, in particular the D-form, and from renewable carbon sources, is thus of great interest.

In recent years, the microbial production of L-LA has gained widespread interest from large biotechnology companies, with several established businesses, including NatureWorks, Purac, Galactic, and several Chinese firms, already producing and supplying L-LA at the half-a-million-tonne scale annually. In comparison, the microbial production of D-LA is less established, with only a handful of published reports [4]. Therefore, the development of an efficient bio-based production platform for D-LA with high optical purity is urgently needed.

Several wild-type bacterial strains can produce D-LA at high titers [5]. These include bacteria in the *Sporolactobacillus* genus (*Sporolactobacillus inulinus*, *S. laevolacticus*, and *S. terrae*) and in the *Lactobacillus* genus (*Lactobacillus delbrueckii* subsp. *delbrueckii*, *L. coryniformis* subsp*. torquens*, and *L. delbrueckii* subsp.). Importantly, several industrial hosts that lack the innate ability to produce D-LA or only accumulate a small amount of D-LA have been engineered to produce D-LA at high titers. In one notable example, Baek and coworkers [6] employed both the heterologous expression of a D-LA-specific enzyme and the deletion of multiple endogenous enzymes to obtain a strain that is able to produce D-LA at a titer of 48.9 g/L (yield 0.79 g/g glucose or 79% of the theoretical yield); however, the productivity rate was still relatively low (0.41 g/L/h) compared with L-LA-producing strains.

*Saccharomyces cerevisiae* is one of the most prominent host organisms for the production of chemicals due to its acidic pH tolerance, high operational stability, and availability of genetic tools [7,8,9,10]. Many approaches have been attempted to increase the productivity of LA in lactate dehydrogenase (LDH) harboring *S. cerevisiae*, aiming to shift the metabolic flux away from ethanol to LA production. These strategies include (1) disrupting the ethanol production pathway by the deletion of specific genes-encoded alcohol dehydrogenase and pyruvate carboxylase, including *adh1*, *pdc1*, *pdc5*, and *pdc6* [11,12,13,14,15]; (2) optimizing lactate dehydrogenase (*ldh*) gene expression [16,17,18]; (3) increasing the availability of NADH by intracellular redox balancing [18]; (4) deleting glycerol-3-phosphate dehydrogenase genes (*gpd1* and *gpd2*) to minimize glycerol production [19]; and (5) reducing ATP consumption by replacing the native pathway for acetyl-CoA production from acetaldehyde with an ATP-independent variant [20,21].

Early efforts in microbial strain developments for producing LA have largely focused on converting simple sugars such as glucose into LA. However, sugars derived from food crops are cost-prohibitive, resource-intensive, and ultimately not an ideal carbon source, especially when considering that demand for food and water is expected to increase significantly ([22], https://en.unesco.org/themes/water-security/wwap/wwdr/2020 (accessed on 14 July 2022)). Therefore, alternative carbon sources, in particular lignocellulosic feedstocks derived from energy crops or agricultural wastes that do not compete for land and water, are desirable. Research into methods for deconstructing lignocellulosic biomass feedstocks into simple sugars as well as the development of alternative carbon sources beyond sugars is a highly active area of study [23]. A major concern when using lignocellulosic biomass feedstocks is the generation of weak acids and other inhibitors during pretreatment and hydrolysis. Unlike lactic acid bacteria, *S. cerevisiae* exhibits a high tolerance to weak acids and other inhibitors and can withstand harsh fermentation conditions, which make it a good candidate for D-LA production from cellulosic hydrolysates. However, D-LA production from cellulosic biomass using wild-type and engineered *S. cerevisiae* has not been studied extensively so far [24].

In this study, we combined rational engineering and yeast mating to create an intraspecific hybrid *S. cerevisiae* strain that is able to produce D-LA efficiently from sugarcane bagasse hydrolysates. Combining these strategies provided an effective way to confer two distinct beneficial traits: weak acid tolerance (from the natural strain) and D-LA production (from the engineered strain). This could not have been achieved using the individual techniques alone.

## 2. Materials and Methods

### 2.1. Strains, Cultivation Conditions, and Reagents

*S. cerevisiae* CEN.PK2-1C strain (*MATa*; *his3D1*; *leu2-3_112*; *ura3-52*; *trp1-289*; *MAL2-8c*; *SUC2*) was obtained from EUROSCARF, Frankfurt, Germany, and *S. cerevisiae* wild-type strain BCC39850 strain was obtained from the Thailand Bioresource Research Center (https://www.tbrcnetwork.org (accessed on 14 July 2022)). *Escherichia coli* DH5α (Invitrogen, Carlsbad, CA, USA) was used for cloning of plasmid DNA. Yeast strains were grown in YPD liquid medium (20 g/L peptone, 20 g/L glucose, and 10 g/L yeast extract) and stored in YPD broth containing 20% glycerol at −70 °C. *E. coli* was grown in Luria-Bertani (LB) medium (10 g/L tryptone, 5 g/L yeast extract, 10 g/L NaCl) and stored in LB broth containing 20% glycerol at −70 °C. Yeast cells were transformed using the LiAc/SS Carrier DNA/PEG method as described previously [25]. Synthetic complete medium (SC) containing 6.7 g/L Yeast Nitrogen Base (BD, Franklin Lakes, NJ, USA), 20 g/L glucose with amino acid dropout was used for selection of *S. cerevisiae* transformants. The CRISPR-Cas9 and CRISPR-Cpf1 (also known as CRISPR-Cas12a) plasmids used in this study for genome editing were generated from pRPR1-gRNA handle-RPR1t [26], p414-TEF1p-Cas9-CYC1t [27], and pUDC175 [28]. All *S. cerevisiae* strains used in this study are summarized in Table 1. All primers used in this study are listed in Appendix A.

### 2.2. Plasmid Construction

The TDH3 promoter and CYC1 terminator were PCR-amplified from genomic DNA of *S. cerevisiae* (strain CEN.PK2-1C) using primer pairs DeltaUp_KpnI_TDH3pro_F and TDH3Pro_R, and CYC1ter_F and CYC1ter_R, respectively. The *S. cerevisiae* codon-optimized *Lm.ldhA* gene (GenBank accession number MW574957) was purchased from Genscript as a plasmid (pUC57-*Lm.ldhA*). *Lm.ldhA* and *HIS3* selectable markers were PCR-amplified from pUC57-*Lm.ldhA* and pRSII405 by using the primer pairs *Lm*LDH_F and *Lm*LDH_R, and CYC1_*Bam*HI_HIS3pro_F and DeltaDown_*Eco*RI_HIS3ter_R, respectively. These four DNA fragments were assembled into a single fragment by overlap-extension PCR (OE-PCR). The resulting cassette was ligated to pJET1.2 vector (Thermo Fisher Scientific, Waltham, MA, USA) to yield pJET-DeltaUp-LmLDH-His3-DeltaDown. To construct pRPR1-gRNA-delta for genome integration, the gRNA-delta fragment was PCR-amplified from pRPR1-gRNA handle-RPR1t using primers delta_gRNA1_*Hin*dIII_F and gRNA_Rev. The 0.13-kb PCR amplicon was gel-purified and ligated via the *Hin*dIII/XhoI sites of pRPR1-gRNA handle-RPR1t to yield pRPR1-gRNA-delta. To construct pArray plasmids for specific gene deletion, the *gpd*1, 2, and *adh*1 specific crRNA for each gene deletion was designed using the CRISPR RGEN tools [29], shown in Appendix A. The crRNA array, which comprises gene-specific crRNA for each target gene as well as direct repeats (5′-AATTTCTACTGTTGTAGAT-3′) and flanking sequences homologous to plasmid backbone pUDE735, was synthesized by GenScript and amplified by using primers crRNA-F and crRNA-R. The plasmid backbone pUDE735 was prepared by PCR amplification of pUDE735 using the primers tSUP4-F and pCAS9-R. Homologous recombination of the synthesized fragment with the plasmid backbone pUDE735 resulted in pArray-GPD1,2 plasmid for *gpd1* and *gpd2* gene deletion and pArray-ADH1 plasmid for *adh1* gene deletion.

### 2.3. Yeast Strain Construction

The CRISPR/Cas9 system was employed to create the CEN.PK2-1C_DLDH strain. The *Lm*LDH expression cassette donor DNA was PCR-amplified from plasmid pJET-DeltaUp-*Lm*LDH-His3-DeltaDown using DeltaUp_TDH3pro_F and DeltaDown_HIS3ter_R primers and then transformed into competent CEN.PK2-1C cells along with p414-TEF1p-Cas9-CYC1t [27] and pRPR1-delta-gRNA-RPR1t [26], which harbored the SpCas9 gene and a single-guide RNA targeting the Ty retrotransposon delta sites, respectively. Transformants were selected on SC medium with L-histidine/L-tryptophan/uracil dropout. Colony PCR of transformants was performed using the DeltaUp_F and LmLDH_int_seq_R primers.

### 2.4. CRISPR-Cpf1-Mediated Gene Deletion

The markerless CRISPR/Cas12a system was employed to generate deletion strains. Donor DNA for each gene deletion was obtained by PCR using complementary primers. The donor DNA was gel purified and then transformed into competent CEN.PK2-1C_DLDH cells along with the corresponding pArray plasmid and pUDC175, which contains the *Cpf1* (Cas12a) gene from *Francisella tularensis* under the control of the TEF1 promoter. Transformants were selected on SC medium with L-histidine/L-tryptophan dropout containing 200 µg/mL G418. Colony PCR using colony PCR primers were performed to verify the gene deletions.

### 2.5. Yeast Mating

The wild-type *S. cerevisiae* BCC39850 (*MATalpha*; weak acid-tolerant) and CEN.PK2-1C_DLDHΔ*gpd* (*MATa*) strains were crossed in YPD agar plate and grown overnight at 30 °C. After mating, a loop full of the mated population was suspended in SC broth and streaked on SC agar plate (without amino acid), and then incubated at 30 °C for 3–5 days until diploid colonies appeared. Diploid colonies were taken and re-streaked on sporulation agar plates (10 g/L potassium acetate, 1 g/L yeast extract, 0.5 g/L dextrose, 20 g/L agar) and incubated at 30 °C for 3–5 days to confirm tetrad formation under microscope.

### 2.6. Batch Fermentation, Growth Curve, and Lactic Acid Production under Non-Neutralized Conditions

D-LA fermentation was carried out in 250 mL Erlenmeyer flasks containing 50 mL of YPD containing 100 g/L glucose with *S. cerevisiae* cells suspended at an initial density OD600 of 0.05. The synthetic inhibitor cocktail was formulated to mimic the composition of hydrolysate toxins present in sugarcane bagasse from the alkaline pretreatment process as modified from van der Pol and coworkers [30]. To evaluate the production of D-LA in the presence of synthetic hydrolysate toxins, an equal volume of synthetic hydrolysate toxins was added to 50 mL of YPD (100 g/L glucose), resulting in a medium containing 0.35 g/L formic acid, 1.65 g/L acetic acid, 0.025 g/L levulinic acid, 0.047 g/L vanillin, and 0.018 g/L syringaldehyde with the same initial OD_600_ of *S. cerevisiae* cells. The cultures were incubated at 30 °C with rotary shaking at 200 rpm. For time-course experiments, samples were taken every 8 h for D-LA production analysis.

### 2.7. Alkaline Pretreatment of Sugarcane Bagasse and Simultaneous Saccharification and Fermentation

Sixty grams of sugarcane bagasse was pretreated by adding 180 mL of 50 g/L NaOH and heated at 90 °C for 90 min under air atmosphere. The whole slurry of the pretreated sugarcane bagasse was prepared by adding 400 mL of water, and pH was adjusted to neutral with sulfuric acid. The total volume was made up to 1 L to achieve the final concentration of 6% loading of whole slurry. The washed solid was also obtained from the same pretreatment condition, in which the solid fraction was separated and washed with tap water until neutral pH was reached and the solid sample was then dried at 50 °C. Chemical compositions of the pretreated biomass samples were analyzed according to National Renewable Energy Laboratory (NREL) analytical procedures [31]. The composition of washed solid was reported as 55.1% cellulose, 18.1% hemicellulose, and 8.7% lignin. To evaluate the D-LA production efficiency of *S. cerevisiae* strains, the washed solid and whole slurry of alkaline-pretreated sugarcane bagasse (6% solid loading) were used as the carbon substrates for simultaneous saccharification and fermentation (SSF). YP medium (10 g/L yeast extract and 20 g/L peptone) containing sugarcane bagasse solid fraction or whole slurry supplemented with 30 filter paper units (FPU)/g biomass commercial cellulase (Cellic Ctec2; Novozyme Inc., Franklinton, NC, USA) was inoculated with *S. cerevisiae* cells at OD_600_ of 0.5 and incubated at 30 °C for 48 h with shaking at 200 rpm. The amounts of sugar degradation byproducts, including carboxylic acids, furfural, and hydroxymethyl furfural present in the whole slurry are shown in Appendix A.

### 2.8. Product Analysis

Samples from both batch fermentation and simultaneous saccharification and fermentation were harvested by centrifugation (10,000× *g* for 5 min at 4 °C) and filtered through 0.2-micron filter cellulose acetate membranes (Millipore; Milford, MA, USA). The yields of sugars, lactic acid, ethanol, and glycerol were determined using a high-performance liquid chromatographic (HPLC) system (Shimadzu Prominence LC-20 equipped with a refractive index detector (Shimadzu Corporation, Kyoto, Japan) and an Aminex-HPX-87H Column (Bio-Rad, Hercules, CA, USA)). The column temperature was maintained at 65 °C and 5 mM H_2_SO_4_ was used as the mobile phase at a flow rate of 0.5 mL/min.

## 3. Results

### 3.1. Engineering S. cerevisiae to Produce D-Lactic Acid

To engineer the yeast *S. cerevisiae* to produce high titers of D-LA, we first introduced the D-lactate dehydrogenase (D-LDH) gene from *Leuconostoc mesenteroides*, a natural D-LA producer, into the laboratory strain CEN.PK2-1C. To increase the number of chromosomally integrated D-LDH expression cassettes, we employed a CRISPR-based strategy developed by Shi and coworkers [32], whereby the *Lm.ldhA* gene construct of interest is integrated into multiple Ty retrotransposon delta sites spread throughout the yeast genome. The resulting strain, named CEN.PK2_DLDH, was able to produce D-LA at a titer of 2.3 ± 0.2 g/L after three days of fermentation in 50 mL Falcon tubes (Figure 1). In addition to the desired product, D-LA, the strain also produced ethanol as a co-product (1.7 ± 0.2 g/L) and a small amount of glycerol (0.042 ± 0.01 g/L). To increase the production level of D-LA and minimize the production of the side-products ethanol and glycerol, we targeted *gpd1*, *gpd2*, and *adh1* for deletion by employing the CRISPR/Cas12a (previously called *Cpf1*) system. We observed the highest D-LA titer in the triple deletion strain (CEN.PK2_DLDHΔ*gpd*Δ*adh1*), which produced D-LA at a titer of 14.4 ± 1.8 g/L, a 6.3-fold titer improvement over the level observed in strain CEN.PK2_DLDH. Similarly, we saw improvement in the double deletion strain (CEN.PK2_DLDHΔ*gpd*), which produced D-LA at a titer of 9.6 g/L ± 0.4 g/L. The triple deletion strain also produced less ethanol (0.5 ± 0.1 g/L) and an undetectable level of glycerol. However, the improvements in the D-LA titer and yield in the triple deletion strain were accompanied by severe growth retardation (Figure 1D) and a reduction in the maximum specific growth rate (Appendix A). Given the unfavorable growth phenotype of the triple deletion strain, we chose the double deletion strain CEN.PK2_DLDHΔ*gpd* for further improvement by yeast breeding.

### 3.2. Improving D-LA Production of CEN.PK2_DLDHΔgpd by Conventional Yeast Mating

The double deletion knockout strain expressing D-LDH (CEN.PK2_DLDHΔ*gpd*) was crossed with the weak acid-tolerant strain *S. cerevisiae* BCC39850 (Appendix A). Over two-hundred diploid progenies were screened for D-LA production before choosing the representative hybrid strains for further analysis. The presence of an integrated *Lm.ldhA* gene in the three selected putative progeny strains (hybrid2, hybrid35, and hybrid36) and the parental strain CEN.PK2_DLDHΔ*gpd* was demonstrated by PCR (Figure 2A). The status of the *gpd* genes was assessed by PCR using the *gpd1* and *gpd2* primers, which revealed the presence of both the intact and deleted copies of *gpd1* and *gpd2* in hybrid2 (Figure 2B). Hybrid 35 contained intact *gpd1* and deleted *gpd2*, whereas intact *gpd1* and *gpd2* genes were found in hybrid36 (Figure 2B). The selected hybrid strains were also evaluated for lactic acid tolerance by spotting on YPD plates containing 0, 30, 40, 50, and 60 g/L of DL-lactic acid (DL-LA). Growth was observed at day 1 and day 3 after incubating at 30 °C (Figure 3). The growth of the double knockout CEN.PK2_DLDHΔ*gpd* parental strain was markedly slower than hybrid2 and hybrid35 on unsupplemented YPD. In the presence of 40–50 g/L of DL-LA, no growth was observed for CEN.PK2_DLDHΔ*gpd*, whereas the other strains still showed a varying degree of growth. Based on the growth phenotype, hybrid2 was considered to be the most tolerant strain to DL-LA as it was the only one able to overcome a DL-LA concentration of 50 g/L (Figure 3). The *Lm.ldhA* copy number among the strains was also quantified by qPCR (Appendix A). The *Lm.ldhA* copy number of hybrid2 was only 0.93 while hybrid35 and hybrid36 contained about 2.98, which could result from the different patterns of meiotic segregation. Next, the D-LA productivity in each strain was assessed under aerobic fermentation conditions in YPD broth with or without synthetic hydrolysate toxins.

### 3.3. Batch Fermentation for Lactic Acid Production under Non-Neutralized Conditions

The fermentative abilities of CEN.PK2_DLDHΔ*gpd*, hybrid2, hybrid35, and hybrid36 were analyzed in the YPD medium containing 100 g/L glucose with or without synthetic toxins. The glucose consumption and D-LA and ethanol production (as byproduct) were monitored at 8 h intervals of cultivation time. In both fermentation conditions, no neutralizing agents were added. Figure 4 shows the glucose consumption and the D-LA and ethanol production. In the absence of synthetic hydrolysate toxins, hybrid2 consumed glucose the fastest, in which 50% of the glucose was consumed after 8 h. The glucose consumption was lower for hybrid35 and hybrid36, and CEN.PK2_DLDHΔ*gpd* demonstrated the slowest consumption (Figure 4C). The maximum production of D-LA and ethanol was measured (Figure 4B). The highest D-LA production was obtained with hybrid36. The CEN.PK2_DLDHΔ*gpd* strain produced the least D-LA but still produced almost as much ethanol as hybrid35. In addition to D-LA and ethanol, some glycerol was produced in fermentations with hybrid2, hybrid35, and hybrid36 (Appendix A). No glycerol product was detected in the fermentations with CEN.PK2_DLDHΔ*gpd*.

The effects of synthetic hydrolysate toxins on microbial growth, glucose consumption, and the production of D-LA and ethanol were also investigated in aerobic batch fermentation. Overall, the addition of the synthetic hydrolysate toxins leads to an initial delay in the glucose uptake (Figure 4F) followed by reduced consumption. However, there are differences as hybrid35, hybrid36, and CEN.PK2_DLDHΔ*gpd* exhibited longer lag times than hybrid2 (Figure 4F).

Theoretically, one gram of D-LA can be produced from one gram of glucose catalyzed by D-lactate dehydrogenase (DLDH). Overall, the values of the D-LA productivity and yield of hybrid2, hybrid35, and hybrid36 were markedly greater than that of the parental strain CEN.PK2_DLDHΔ*gpd* (Table 2). When the synthetic hydrolysate cocktail was added, all the strains exhibited the reduced glucose uptake and a reduction in the growth rate and D-LA productivity. The highest D-LA productivity was observed for hybrid2 under both conditions. Similarly, it was found that synthetic hydrolysate toxins also affected the productivity of ethanol in the same way as D-LA productivity. A marked reduction in ethanol productivity was observed in both hybrid35 and hybrid36, whereas there was little effect on ethanol productivity in CEN.PK2_DLDHΔ*gpd* and hybrid2.

### 3.4. D-Lactic Acid Production Using Alkaline-Pretreated Sugarcane Bagasse in SSF

The D-LA production from alkaline-pretreated bagasse was investigated for the strain with the highest productivity (hybrid2) and the parental strain CEN.PK2_DLDHΔ*gpd* as the control (Figure 5). The SSF of the washed solid fraction (Figure 5A) showed that the maximum D-LA production was higher for hybrid2. In contrast, CEN.PK2_DLDHΔ*gpd* produced more ethanol than hybrid2. No glucose was detected during the SSF, indicating efficient sugar utilization by both strains during the SSF of the washed solid. The glucose consumption of CEN.PK2_DLDHΔ*gpd* during the SSF of the whole slurry was quite limited which directly affected both D-LA and ethanol production (Figure 5B). Hybrid2 was able to consume glucose more efficiently and produced more D-LA than CEN.PK2_DLDHΔ*gpd* during the SSF of the whole slurry. Alkaline pretreatment at high temperature can also release a large amount of acetic acid and phenolic compounds into the liquid fraction of the whole slurry [30]. This could inhibit either microbial growth or activities of enzymes that are used for saccharification [33]. The D-LA productivity was greater from the washed solid compared with the whole slurry (Table 3). Because there were no inhibitor byproducts and fermentable sugars detected in the sugarcane bagasse washed solid fraction, almost all of the washed solid fraction was gradually converted to glucose (total 32.7 ± 0.43 g/L) and xylose (12.21 ± 0.16 g/L). The analysis of the degradation byproducts in the whole slurry showed that the most abundant inhibitor was acetic acid (3.06 ± 0.15 g/L), whereas formic acid, levulinic acid, furfural, and hydroxymethy furfural were not found (Appendix A).

## 4. Discussion

In this study, a transgenic strain of *S. cerevisiae* was established that expresses *L. mesenteroides D-LDH* genes integrated into the genome. D-LA was produced from this strain as expected, although the co-production of ethanol and glycerol limited the D-LA productivity. The disruption of the *gpd1* and *gpd2* genes in this strain would eliminate glycerol-3-phosphate dehydrogenase activity and thus the production of glycerol. In *S. cerevisiae*, Gpd1 and Gpd2 are the rate-controlling enzymes for glycerol synthesis. Both isoforms play important roles in osmotic adaptation, the response to oxidative stress, heat shock protection, and redox balance [34]. Glycerol also participates in the biosynthesis of glycerophospholipids and triacylglycerols by conversion to the intermediate G3P and plays a role in maintaining cytosolic redox balance, all of which are important for *S. cerevisiae* to retain normal physiological functions and growth [35]. The deletion of the *gpd1* and *gpd2* genes in *S. cerevisiae* might be beneficial for the production of high-value chemicals because glycerol is considered to be a major byproduct, accounting for 5% of carbon during cell growth. However, the deletion of both *gpd* genes had a negative impact on growth (Figure 1D). Nissen and coworkers [36] reported a similar finding, in which a double deletion *gpd* mutant showed a 29% reduction in biomass synthesis with a 12.7% increased ethanol yield, whereas single deletion mutants (*gpd1*Δ and *gpd2*Δ) showed very little increased ethanol yield under aerobic conditions. Furthermore, the volumetric productivity of ethanol by *S. cerevisiae* can be increased by fine-tuning GPD expression during fed-batch fermentation [35].

We hypothesized that D-LA production and other fermentation characteristics could be improved by mating the CEN.PK2_DLDHΔ*gpd* strain with a wild-type strain. Three progeny strains were isolated. Hybrid35 had a deletion of *gdp2* whereas hybrid36 appeared to be wild-type with respect to both *gpd* genes. Hybrid2 appeared to be the diploid hybrid, as shown by the presence of wild-type and deleted copies of both *gdp* genes (Figure 2B). The superior growth and D-LA production characteristics of hybrid2 could be attributed to heterosis, although it is not known which genetic factor(s) from the BCC39850 parent contribute to the superior characteristics of hybrid2 compared with those of the CEN.PK2_DLDHΔ*gpd* parent. The extended lag phase corresponded to a detoxification phase in which *S. cerevisiae* adapts and responds to the inhibitors [37]. These affected the ethanol and D-LA productivity especially in hybrid35 and hybrid36. Because the genetic backgrounds of CEN.PK2-1C and BCC39850 are different, the hybrid strains constructed in this study responded to inhibitors in different manners. The physiological and metabolic changes, such as the composition of plasma membrane rearrangements, the ability to maintain intracellular pH homeostasis, the detoxification of reactive oxygen species, and the assimilation of inhibitors, are reported to the improved yeast strains [37,38].

Compared with previous reports on LA production in engineered *S. cerevisiae* (Table 4), the D-LA productivity of hybrid2 was higher except for the strain reported in [39]. Moreover, the D-LA yield of hybrid2 is in the same range as that reported for other *S. cerevisiae* strains with disruption of ADH and PDC genes [6,19,39,40] in which the metabolic flux was shifted toward LA production. The approach of crossing genetically engineered strains expressing heterologous DLDH with wild-type could be applied for the further improvement of D-LA production yield, for example, in engineered strains with deletions of other genes, including ADH, PDC, and D-Lactate dehydrogenase gene (*dld*1).

The D-LA production from a cellulosic biomass was also demonstrated using the CEN.PK2_DLDH Δ*gpd* and hybrid2 strains. Alkaline-pretreated sugarcane bagasse was used as a representative feedstock for D-LA production by the SSF to avoid substrate inhibition that could potentially inhibit microbial fermentation and enzymatic saccharification. The major byproduct found in the whole slurry was acetic acid. To cope with acid stress, *S. cerevisiae* responds to the changes of the intracellular pH from the D-LA and the external acetate/H+ in ways that utilize a large amount of ATP, which leaves less ATP available for cell growth [41]. As a result, the D-LA productivity by hybrid2 from the whole slurry was lower than that obtained from the washed solid fraction.

To our knowledge, there are no other reports of D-LA production from sugarcane bagasse using engineered *S. cerevisiae*. The D-LA productivity obtained from hybrid2 was comparable to that reported for the fermentation of corn stover hydrolysate using lactic acid bacteria (LABs), which ranged from 0.32 to 1.02 g/L/h [1]. In terms of industrial applicability, D-LA production in S. cerevisiae has advantages over LABs, including the ability to cope with environmental stresses, low pH, and no requirement of complex nutrition for growth [10]. With the advantages of sustainable, environmental, and socio-economic aspects as well as cost-competitiveness, the use of lignocellulosic feedstocks for D-LA production needs to be studied more intensively. Here, we have shown that it is feasible to produce D-LA from sugarcane bagasse, an economically important lignocellulosic feedstock, using engineered yeast.

## 5. Conclusions

In this study, *S. cerevisiae* expressing heterologous *Lm*D-LDH (CEN.PK2_DLDH) was constructed using a CRISPR-Cas9-based strategy. To minimize ethanol and glycerol side-products, and thus consequently increase the production of D-LA, we deleted the *gpd1*, *gpd2*, and *adh1* genes by employing the CRISPR/Cas12a system. This strategy was successful, as shown by the reduction in side-products and increased D-LA production for both the double deletion CEN.PK2_DLDHΔ*gpd* and triple deletion CEN.PK2_DLDHΔ*gpd*Δ*adh1* strains. However, the improvements in the D-LA titer and yield were accompanied by growth retardation, which was pronounced for the triple deletion strain. Therefore, the double deletion CEN.PK2_DLDHΔ*gpd* strain was crossed with the weak acid-tolerant *S. cerevisiae* BCC39850 wild-type strain for strain improvement. The D-LA productivity and yield of the isolated hybrid2 were significantly greater under the fermentation conditions containing formulated hydrolysate inhibitors compared with CEN.PK2_DLDHΔ*gpd*. Moreover, hybrid2 also showed superior D-LA production in simultaneous saccharification and fermentation using alkaline-pretreated sugarcane bagasse on washed solid and whole slurry. The work demonstrates the use of combined conventional breeding and gene editing for developing a yeast cell factory with improved performance for D-LA production from cellulosic substrates with industrial potential.

## Figures and Tables

**Figure 1 jof-08-00816-f001:**
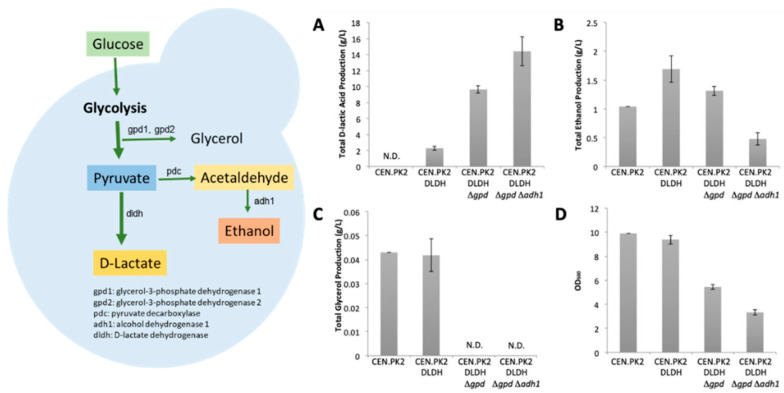
Metabolic and growth profiles of engineered strains. (**A**) Production of D-lactic acid, (**B**) ethanol, (**C**) glycerol, and (**D**) growth profile of the engineered strains. The cultures were grown at 30 °C and 250 rpm in an orbital shaking incubator. Samples were taken after 3 days, and the supernatants were analyzed on HPLC to quantify the content of each metabolite (N.D. indicates that the metabolite was not detected). Data represent mean values of biological triplicates and error bars indicate standard deviations.

**Figure 2 jof-08-00816-f002:**
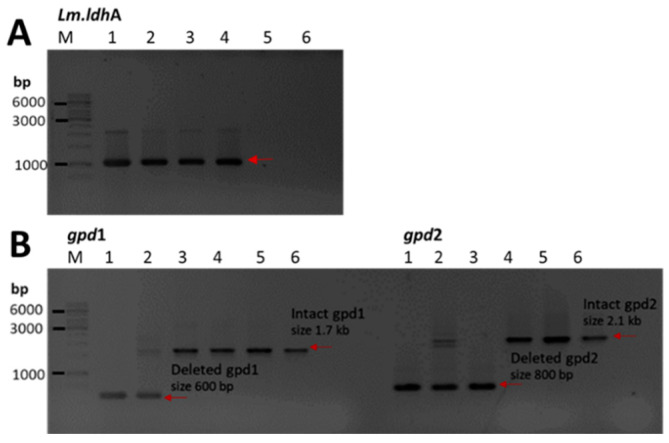
Construction of *S. cerevisiae* strains from yeast mating. (**A**) PCR amplification of *Lm.ldhA* gene (1kb) from the genomic DNA of the strain CEN.PK2_DLDHΔ*gpd* (lane 1), hybrid2 (lane 2), hybrid35 (lane 3), hybrid36 (lane 4), wild-type laboratory strain CEN.PK2-1C (lane 5), and wild isolate BCC39850 (lane 6). (**B**) PCR amplification of *gpd* genes (*gpd1* and *gpd2*; indicated above the lanes) from the genomic DNA of the strain CEN.PK2_DLDHΔ*gpd* (lane 1), hybrid2 (lane 2), hybrid35 (lane 3), hybrid36 (lane 4), wild-type laboratory strain CEN.PK2-1C (lane 5), and wild isolate BCC39850 (lane 6). The products from PCR amplification were separated by agarose gel electrophoresis and stained with ethidium bromide. Sizes of PCR amplicons were estimated by comparison with DNA markers (lanes marked M), in which fragments of known length are indicated on the left.

**Figure 3 jof-08-00816-f003:**
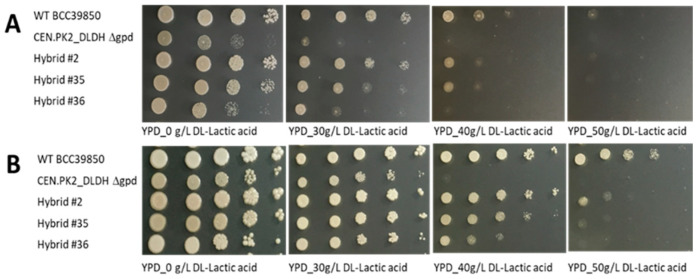
Lactic acid tolerance of *S. cerevisiae* BCC39850 (wild-type parent), CEN.PK2_DLDHΔ*gpd* (transgenic parental strain), and progeny hybrid2, hybrid35, and hybrid36. Strains were grown in liquid culture to an initial yeast inoculum OD_600_ of 0.5 before spotting on plates. (**A**) Growth at day 1 after spotting on a YPD plate containing 0, 30, 40, 50, and 60 g/L DL-lactic acid and incubation at 30 °C. (**B**). Growth at day 3 after spotting on a YPD plate containing 0, 30, 40, 50, and 60 g/L DL-lactic acid and incubation at 30 °C.

**Figure 4 jof-08-00816-f004:**
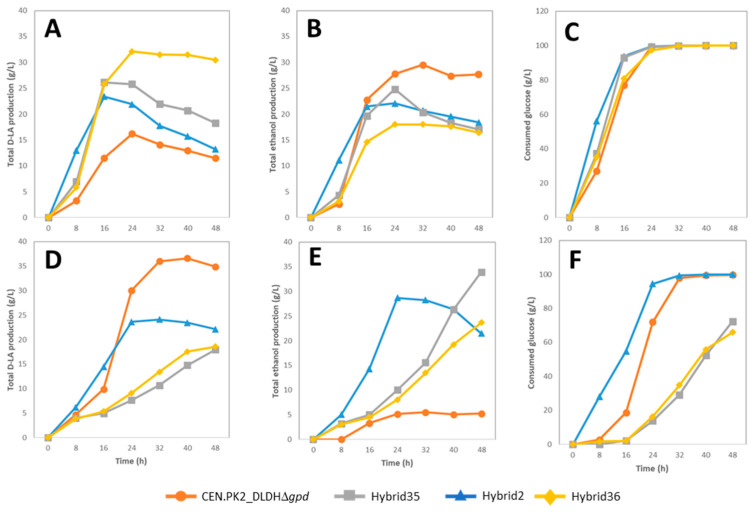
Time courses of glucose conversion to D-lactic acid and ethanol under aerobic conditions. D-lactic acid production in YPD medium containing 100 g/L glucose without synthetic hydrolysate toxins (**A**); ethanol production (**B**) and consumed glucose (**C**). D-lactic acid production in YPD medium containing 100 g/L glucose with synthetic hydrolysate toxins (0.36 g/L formic acid, 1.65 g/L acetic acid, 0.025 g/L levulinic acid, 0.05 g/L vanillin, and 0.018 g/L syringaldehyde) (**D**); ethanol production (**E**) and consumed glucose (**F**). Data represent mean values of biological triplicates and error bars indicate standard deviations.

**Figure 5 jof-08-00816-f005:**
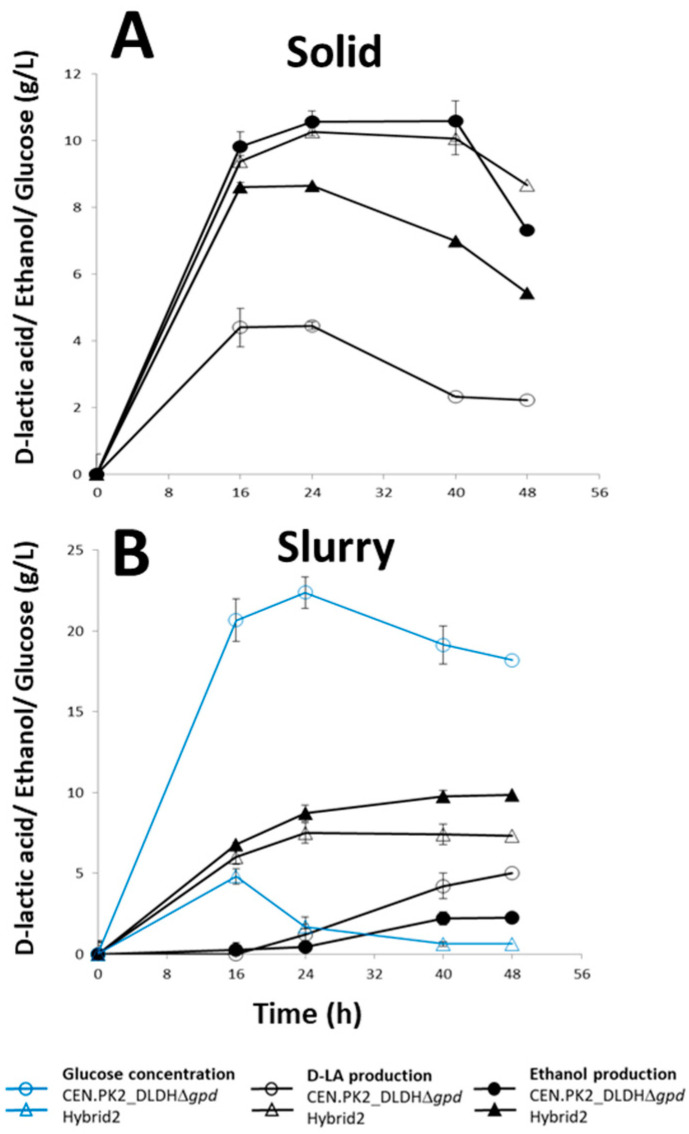
Simultaneous saccharification and fermentation (SSF) of D-lactic acid and ethanol by CEN.PK2_DLDHΔ*gpd* and hybrid2: (**A**) SSF of washed solid fraction of alkaline-pretreated sugarcane bagasse supplemented with Cellic^®^ Ctec2 at 30 FPU/g biomass. (**B**) SSF of whole slurry of alkaline-pretreated biomass (6% solid loading) supplemented with Cellic^®^ Ctec2 at 30 FPU/g biomass. Glucose concentration in SSF medium is shown by the solid blue line; parental strain CEN.PK2_DLDHΔ*gpd* (open circle), hybrid2 (open triangle). Production of D-lactic acid is shown by the solid black line; parental strain CEN.PK2_DLDHΔ*gpd* (open circle), hybrid2 (open triangle). Production of ethanol is shown by the solid black line; CEN.PK2_DLDHΔ*gpd* (closed circle), hybrid2 (closed triangle). All SSF experiments were performed in triplicate at the temperature of 30 °C with shaking at 200 rpm.

**Table 1 jof-08-00816-t001:** Strains used in this study.

Strain Name	Genotype	Description	Reference
CEN.PK2-1C	*MATa*; *his3D1*; *leu2**-**3_112*; *ura3**-**52*; *trp1**-**289*; *MAL2**-**8c*; *SUC2*	Laboratory strain	Euroscarf
BCC39850	*MATalpha*	Wild type; tolerant to weak acids	This study
CEN.PK2_DLDH	CEN.PK2-1C P_TDH3_*-**Lm*.*ldhA*	CEN.PK2-1C overexpressing *S*. *cerevisiae* codon-optimized *ldh*A from *Leuconostoc mesenteroides*	This study
CEN.PK2_DLDHΔ*gpd* Δ*adh1*	CEN.PK2*_DLDH* Δ*gpd1* Δ*gpd2*Δ*adh1*	CEN.PK2_DLDH with *gpd1 gpd2* and *adh1* deleted	This study
CEN.PK2_DLDHΔ*gpd*	CEN.PK2*_DLDH* Δ*gpd1* Δ*gpd2*	CEN.PK2_DLDH with *gpd1 gpd2* deleted	This study
Hybrid2	*MATa**/**alpha* P_TDH3_*-**Lm*.*ldhA* Δ*gpd1* Δ*gpd2* clone 2	Hybrid strain between the haploid of BCC39850 and CEN.PK2_DLDH Δ*gpd*; clone 2	This study
Hybrid35	*MATa**/**alpha* P_TDH3_*-**Lm*.*ldhA* Δ*gpd1* Δ*gpd2* clone 35	Hybrid strain between the haploid of BCC39850 and CEN.PK2_DLDH Δ*gpd*; clone 35	This study
Hybrid36	*MATa**/**alpha* P_TDH3_*-**Lm*.*ldhA* Δ*gpd1* Δ*gpd2* clone 36	Hybrid strain between the haploid of BCC39850 and CEN.PK2_DLDH Δ*gpd*; clone 36	This study

**Table 2 jof-08-00816-t002:** D-lactic acid (D-LA) productivity and yield in batch fermentation using glucose as a sole carbon source.

Strain	Productivity (g/L/h)		Yield (g/g)	
	D-LA	Ethanol	D-LA	Ethanol
Without synHT ^1^				
CEN.PK2_DLDHΔ*gpd*	0.70 ± 0.03	1.29 ± 0.11	0.16 ± 0.04	0.29 ± 0.07
Hybrid2	1.54 ± 0.10	1.37 ± 0.05	0.24 ± 0.02	0.21 ± 0.02
Hybrid35	1.35 ± 0.08	1.13 ± 0.04	0.27 ± 0.01	0.23 ± 0.02
Hybrid36	1.48 ± 0.03	0.83 ± 0.05	0.32 ± 0.01	0.18 ± 0.02
With synHT ^1^				
CEN.PK2_DLDHΔ*gpd*	0.19 ± 0.02	1.19 ± 0.03	0.06 ± 0.00	0.39 ± 0.03
Hybrid2	1.05 ± 0.03	0.94 ± 0.02	0.28 ± 0.01	0.25 ± 0.00
Hybrid35	0.57 ± 0.02	0.35 ± 0.02	0.50 ± 0.06	0.30 ± 0.02
Hybrid36	0.46 ± 0.09	0.41 ± 0.02	0.36 ± 0.06	0.35 ± 0.06

^1^ Synthetic hydrolysate toxins.

**Table 3 jof-08-00816-t003:** D-lactic acid (D-LA) productivity and yield in SSF of washed solid and whole slurry of alkaline-pretreated sugarcane bagasse.

Strain	Productivity (g/L/h)		Conversion Yield (g/g Glucan)	
	D-LA	Ethanol	D-LA	Ethanol
Washed solid				
CEN.PK2_DLDHΔ*gpd*	0.23 ± 0.05	0.59 ± 0.01	0.14 ± 0.02	0.34 ± 0.01
Hybrid2	0.59 ± 0.01	0.54 ± 0.01	0.33 ± 0.01	0.28 ± 0.00
Whole slurry				
CEN.PK2_DLDHΔ*gpd*	0.11 ± 0.00	0.01 ± 0.01	0.11 ± 0.00	0.06 ± 0.01
Hybrid2	0.34 ± 0.02	0.58 ± 0.01	0.24 ± 0.01	0.26 ± 0.00

**Table 4 jof-08-00816-t004:** Comparison of genetically engineered *Saccharomyces cerevisiae* for lactic acid production.

Strain	Substrate	Productivity (g/L/h)	Yield (g/g)	Titer (g/L)	Reference
*S**. cerevisiae* OC2 (*pdc*1::P*pdc*1-D-LDH (Bovine-LDH))	glucose	1.21	0.65	50.6	[13]
*S**. cerevisiae* OC2 (*pdc*1::P*pdc*1-D-LDH (*L*. *mesenteroides* DLDH))	glucose	0.85	0.61	61.5	[12]
*S**. cerevisiae* SR8 (*Rhizopus oryzae* LDHA)	glucose	1.05	0.17	6.9	[2]
*S**. cerevisiae* SR8L (*Rhizopus oryzae* LDHA)	glucose	1.32	0.22	9.9	[2]
*S**. cerevisiae* JHY5330 (DLDH*dld*1Δ*jen*1Δ*adh1*Δ*gpd*1Δ*gpd*2 Δ*pdc*1Δ)	glucose	0.41	0.79	48.9	[6]
CEN.PK2-1C (*pdc*1Δ*cyb*2Δ *gpd*1ΔP*ccw*12 _LDH*adh*1ΔP*gpd*_mhpFald6Δ P*gpd*_eutE)	glucose	0.95	0.80	34.0	[21]
YPH499/dPdA3-34/DLDH/1-18	glucose	2.80	0.65	60.3	[37]
JHY5730	glucose	1.50	0.83	82.6	[38]
*S**. cerevisiae* hybrid2	glucose	1.54	0.24	23.41	This study
*S**. cerevisiae* hybrid2	sugarcanebagasse	0.59	0.32	10.24	This study

## Data Availability

Not applicable.

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
