# Peer review of "D-Lactic Acid Production from Sugarcane Bagasse by Genetically Engineered *Saccharomyces cerevisiae"

_jof, 2022, doi:10.3390/jof8080816_

Round 1
Reviewer 1 Report
The authors developed novel strategy of gene editing combined with conventional breeding to increase productivity of D-LA. These results would be meaningful for gaining basic knowledge academically and technology development related to the production of industrially potent products in this report. Accordingly, I consider this MS as acceptable to be published in Journal of Fungi. However, in order to be accepted as Journal of Fungi, there are some revisions before publication. Additionally, check the English through manuscript by native speaker.
P6L230-231
“the strain also produced a large amount of ethanol (1.7 ± 0.2 g/L)”
What is the concentration of glucose in the medium used for this experiment? Is there any reason to mention that this strain produced a large amount of ethanol (1.7 g/L is 1.7%)?
P6, Fig. 1
Insert the biosynthetic pathway for glycerol, ethanol and D-LA with Gpd1 and Gpd2 etc.
P7, Fig. 3
Describe the correlation between D-LA productivity and lactic acid tolerance, if any.
P8L302-304
“However, there are differences as hybrid35, hybrid36, and CEN.PK2_DLDHΔgpd exhibited longer lag times than hybrid2 (Figure 4D).”
Describe why these are the case (discussion part).
P8L312-314
“Marked reduction in ethanol productivity was observed in both hybrid35 and hybrid36, whereas there was little effect on ethanol productivity in CEN.PK2_DLDHΔgpd and hybrid2.”
Describe why these are the case (discussion part).
P8, Fig. 4
Increase the resolution of the graph. This graph is difficult for determine what the symbol represents.
Author Response
The authors would like to thank the reviewer for their comments and suggestions.
Please see the attachment.

Reviewer 2 Report
The authors present their work on constructing and analysing yeast strains that are improved for their D lactic acid production.
While the data presented are substantial, the novelty of the work and data seem to be limited. As the authors describe in the introduction there have been numerous papers in the last couple of years on the topic. Even using CRISPR cas technology for this aim and addressing the same genetic pathways has been described before.
It is imminent to further explain the novelty and focus on these aspects.
Some questions remain with respect to the data presented:
The authors use breeding techniques to engineer the strains. This is an important aspect of the manuscript, however, the results of diploidization are not easy to understand: is there an explanation why the hybrids which seem to be selected among 200 diploids for the ability to produce D lactic acid are so different between them? Would it not be expected that all of them have two copies of the respective genes, one in the deleted form, one in the non-deleted form? An explanation for this would be valuable.
Line 364: the authors presume that their original engineered strain is limited in D-LA productivity due to coproduction of ethanol and glycerol. However, the data presented in Table 3 do not support this hypothesis as Hybrid2 has a similar productivity of ethanol as the original strain but a significantly increased D-LA productivity. This needs to be explained.
Fig 4A: this is very difficult to read. Please use either more distinct symbols / colours or separate data in different figures.
Fig 2: the legend seems to be missing (what do the numbers stand for?).
Author Response

(The authors gave the same response as above.)

Reviewer 3 Report
The manuscript by Sornlek and colleagues engineered the S.cerevisae strain for D-Lactic acid production. The authors heterologously expressed the D-lactate dehydrogenase (DLDH) gene from Leuconostoc mesenteroides in S.cerevisae. Further, they knocked out gpd1, gpd2 and adh1 to minimize the byproducts ethanol and glycerol. The authors also mated these strains with weak acid-tolerant S.cerevisae strains and isolated the diploid strains producing maximum D-LA. Overall, the data support the conclusion and the experiments are well controlled. I have a few suggestions for improving the manuscript.
1. Figure 1: As shown in Figure1 the major byproduct is ethanol in comparison to glycerol then it would be interesting to knockout only Adh1 and measure the D-LA production which would lead to a reduction in ethanol and might improve the D-LA yield.
2. Figure 1D: Does the graph represents the total OD600 after 3 days or what was the time point for the growth profile?
3. Figure 2: Figure legends could be expanded a bit. Which lanes represent the parental and hybrid strains?
4. Line 265: The range of DL-LA tested is 40-50 instead of 40-60.
5. Table 4: label D-LA and Ethanol column.
Author Response

(The authors gave the same response as above.)

Reviewer 4 Report
The submitted manuscript is very well written and present important findings for the scientific community.
Conclusions are fully supported by the findings.
I recommend manuscript acceptance.
Author Response
The authors would like to thank the reviewer for their comments and suggestions.
Reviewer 5 Report
Referees Comments
on the manuscript entitled “D-lactic acid production from sugarcane bagasse by genetically engineered Saccharomyces cerevisiae” for Journal of Fungi
The authors presented the results on the biosynthesis of D-lactic acid, a practically valuable compound, from а sugarcane bagasse by genetically engineered Saccharomyces cerevisiae. They integrated the bacterial D-lactate dehydrogenase gene from Leuconostoc mesenteroides into several chromosomal regions of the S. cerevisiae CEN.PK2-1C strain to produce D-lactic acid. To reduce the formation of glycerol in this strain, the gpd1 and gpd2 genes encoding glycerol-3-phosphate dehydrogenase were deleted. The constructed strain was crossed with a wild isolate S. cerevisiae BCC39850 isolate selected for resistance to high concentrations of weak acids. The hybrid strain demonstrated a high level of D-lactic acid production using sugarcane bagasse. The article may be published in the Journal of Fungi, however the minor corrections to the manuscript are required.
Point 1: Page 2, lines 76-77 - Include the names of the enzymes encoded by the given genes.
Point 2: Page 2, line 83: Include links to relevant literature.
Point 3: The purpose of the research should be short and clear. The description of the obtained results should be deleted from the Introduction; they can be moved to the Conclusion section.
Point 4: Page 7 - In the caption to figure 2, give explanations for the designation M, 1,2,3,4.
Point 5: Page 9 – It is recommended to delete Table 2 as it duplicates the data shown in Figure 4.
Point 6: Supplementary file(s), TableS3, TableS4, TableS5 - Specify unit (g/L, h-1)
Author Response

(The authors gave the same response as above.)

Round 2
Reviewer 2 Report
The detailed questions and suggestions were answer to satisfaction. However, the main point made in the first round of reviewing has not been answered or worked on. See here: the novelty of the work and data seem to be limited. As the authors describe in the introduction there have been numerous papers in the last couple of years on the topic. Even using CRISPR cas technology for this aim and addressing the same genetic pathways has been described before. It is imminent to further explain the novelty and focus on these aspects.
Author Response
The authors would like to thank the reviewer for raising this important issue.
Comments and Suggestions for Authors
The detailed questions and suggestions were answer to satisfaction. However, the main point made in the first round of reviewing has not been answered or worked on. See here: the novelty of the work and data seem to be limited. As the authors describe in the introduction there have been numerous papers in the last couple of years on the topic. Even using CRISPR cas technology for this aim and addressing the same genetic pathways has been described before. It is imminent to further explain the novelty and focus on these aspects.
Response:
The authors would like to thank the reviewer for raising this important issue.
We have revised the sentence in the abstract to demonstrate the novelty of this work. “Our findings show the exploitation of natural yeast diversity and the potential strategy of gene editing combined with conventional breeding on improving the performance of S. cerevisiae for the production of industrially potent products.” Page 1; Line 29-31
The following paragraph was added to the introduction to state the purpose and highlight the novelty of the work. “In this study, we combined rational engineering and yeast mating to create an intraspecific hybrid S. cerevisiae strain that is able to produce D-LA efficiently from sugarcane bagasse hydrolysates. Combining these strategies provided an effective way to confer two distinct beneficial traits: weak acid tolerance (from the natural strain) and D-LA production (from the engineered strain). This could not have been achieved using the individual techniques alone. Page 3: Line 101-112).